# Valorisation of Wasted Immature Tomato to Innovative Fermented Functional Foods

**DOI:** 10.3390/foods12071532

**Published:** 2023-04-04

**Authors:** Nelson Pereira, Mahsa Farrokhi, Manuela Vida, Manuela Lageiro, Ana Cristina Ramos, Margarida C. Vieira, Carla Alegria, Elsa M. Gonçalves, Marta Abreu

**Affiliations:** 1INIAV—Instituto Nacional de Investigação Agrária e Veterinária, Unidade de Tecnologia e Inovação, 2780-157 Oeiras, Portugal; nmv.pereira@campus.fct.unl.pt (N.P.);; 2Instituto Superior de Engenharia, Universidade do Algarve, 8005-139 Faro, Portugal; a66845@ualg.pt (M.F.);; 3MED—Mediterranean Institute for Agriculture, Environment and Development, CHANGE—Global and Sustainability Institute, Faculty of Science and Technology, Universidade do Algarve, Campus de Gambelas, 8005-310 Faro, Portugal; 4GeoBioTec—Geobiociências, Geoengenharias e Geotecnologias, FCT-UNL, 2829-516 Caparica, Portugal; 5cE3c—Centre for Ecology, Evolution and Environmental Changes, CHANGE—Global Change and Sustainability Institute, Faculdade de Ciências, Universidade de Lisboa, 1749-016 Lisbon, Portugal; 6LEAF—Linking Landscape, Environment, Agriculture and Food Research Center, Associated Laboratory TERRA, Instituto Superior de Agronomia, Universidade de Lisboa, 1349-017 Lisbon, Portugal

**Keywords:** lactic acid bacteria, starter cultures, lactic fermentation, *Lactiplantibacillus plantarum*, *Weissella paramesenteroides*, probiotic potential ingredient

## Abstract

In this study, the lactic fermentation of immature tomatoes as a tool for food ingredient production was evaluated as a circular economy-oriented alternative for valorising industrial tomatoes that are unsuitable for processing and which have wasted away in large quantities in the field. Two lactic acid bacteria (LAB) were assessed as starter cultures in an immature tomato pulp fermentation to produce functional food ingredients with probiotic potential. The first trial evaluated the probiotic character of *Lactiplantibacillus plantarum* (LAB97, isolated from immature tomato microbiota) and *Weissella paramesenteroides* (C1090, from the INIAV collection) through in vitro gastrointestinal digestion simulation. The results showed that LAB97 and C1090 met the probiotic potential viability criterion by maintaining 6 log_10_ CFU/mL counts after in vitro simulation. The second trial assessed the LAB starters’ fermentative ability. Partially decontaminated (110 °C/2 min) immature tomato pulp was used to prepare the individually inoculated samples (Id: LAB97 and C1090). Non-inoculated samples, both with and without thermal treatment (Id: CTR-TT and CTR-NTT, respectively), were prepared as the controls. Fermentation was undertaken (25 °C, 100 rpm) for 14 days. Throughout storage (0, 24, 48, 72 h, 7, and 14 days), all the samples were tested for LAB and Y&M counts, titratable acidity (TA), solid soluble content (SSC), total phenolic content (TPC), antioxidant capacity (AOx), as well as for organic acids and phenolic profiles, and CIELab colour and sensory evaluation (14th day). The LAB growth reached ca. 9 log_10_ CFU/mL for all samples after 72 h. The LAB97 samples had an earlier and higher acidification rate than the remaining ones, and they were highly correlated to lactic acid increments. The inoculated samples showed a faster and higher decrease rate in their SSC levels when compared to the controls. A nearly two-fold increase (*p* < 0.05) during the fermentation, over time, was observed in all samples’ AOx and TPC (*p* < 0.05, r = 0.93; similar pattern). The LAB97 samples obtained the best sensory acceptance for flavour and overall appreciation scores when compared to the others. In conclusion, the *L. plantarum* LAB97 starter culture was selected as a novel probiotic candidate to obtain a potential probiotic ingredient from immature tomato fruits.

## 1. Introduction

Continuous population expansion and the high estimated food waste along the food chain have caused a demand–supply imbalance with negative impacts on the environment, economy, and on food security. Food waste accounts for one-third of world production [1], and it remains a major global concern with negative impacts on the economy, nutrition, food security, and on the environment. Food loss occurs at all stages of the food supply chain, from agricultural production to the transformation, distribution, and consumption of the food [2]. The fruit and vegetable sector is particularly liable for considerable waste, with estimated losses of ca. 40–50% worldwide [3]. The tomato (*Lycopersicon esculentum* L.) crop is one of the most important in the world. Its annual production was estimated at around 17 million tonnes in Europe (especially in Italy, Spain, and Portugal) and about 182 million tonnes worldwide [4]. Therefore, there is an excellent potential for the valorisation of agricultural biomass in the tomato supply chain, including the fraction that is deemed unsuitable for sale in the fresh fruit market (e.g., due to unacceptable injuries, colour, shape, or maturity), as well as from the side flows from the processing industry [5]. The tomato industry in Portugal, which is of great national importance and high turnover (5th world exporter), is focused on obtaining a single high-value product—tomato paste [6]. Only fruits at the red ripe stage (i.e., when entirely red mature) enter the processing plant. Consequently, in 2015, and according to the Portuguese Competence Centre for Industry Tomato (Centro de Competências para o Tomate de Indústria, Cartaxo, Portugal), an estimated 112 kilotonnes of green tomatoes were left in the Portuguese fields, thus representing an unsustainable use of natural resources (water, soil, and energy) [7]. This pool of discarded fruit represents a unique opportunity by which to produce commercial value-added products [8]. On the other hand, this strategy also allows this industry to diversify products as an innovative trend in the food chain, to create value at the product level, and to efficiently use natural resources.

Food preservation by fermentation is an ancient and widely practised technology, providing foods with enhanced organoleptic properties, nutrient profile, shelf life, and food safety improvements [9]. The exploitation of wasted by-products in producing fermented foods using lactic acid bacteria (LAB) and other microorganisms has been widely researched [7,10]. Additionally, most fermented foods are dairy- and/or animal-based products [11]. However, the rising prevalence of plant-based diets (vegetarianism and veganism) and health conditions, such as lactose intolerance, hypercholesterolemia, and allergies to cow’s milk proteins, have increased consumer demand for novel plant-based alternatives. Accordingly, the food industry has been developing fermented, non-dairy, health-promoting plant-based products to meet the above needs [12].

The fermentation process allows for the reduction in undesirable compounds that might affect the rate and extent of essential bioconversions, increases in product digestibility, and contributions to promoting human health. The fermentation process can occur spontaneously (natural fermentation) or can be triggered by adding starter cultures. Controlled fermentation with the addition of selected strains, unlike natural fermentation, is considered advantageous for the commercial production of fermented plant-based products. Under this approach, it is possible to speed up the fermentation process, reduce the risk of undesirable changes in products’ organoleptic properties (such as off flavours), and to provide uniform quality products [10]. The traditional ways of preserving raw materials have favoured selecting specific bacterial lineages that are well adapted to fermented products [13]. LAB are part of tomato fruit microbiota and many authors have explored the potential use of autochthonous bacterial strains in the fermentation process [9]. The selection of LAB strains to ferment tomatoes has also proved essential by which to develop fermented products with a suitable volatile profile, thereby directly influencing the products’ flavour [14].

Recently, the food industry has also focused on consumers’ demand for products that provide health benefits beyond their nutritional value [15]. Developing probiotic food formulations is another key research area for the future functional food market [16]. Probiotic strains are live microorganisms that are able to adapt and survive in a dynamic environment, as well as are able to confer health benefits to the host when consumed in appropriate amounts in the food [17]. Specifically, the minimum concentration of probiotics is approximately 6–7 log_10_ CFU/mL during food product consumption [16]. Bacteria encounter a variety of stress factors that may hinder their growth when ingesting food, namely acidity and bile salts in the digestive tract, the increased concentration of specific ions or nutrient depletion, and exposures to osmotic and oxidative stress in product matrices. These adverse conditions may detrimentally affect their viability and functionality [17]. Therefore, microbes with probiotic potential must maintain their survival under these challenging conditions [18] and are assessed by in vitro and in vivo tests.

To the best of our knowledge, none of the studies aimed at valorising unripe tomatoes through controlled fermentation has been researched by using starters with probiotic potential. This study aimed to develop a high-value, potential probiotic ingredient by the controlled fermentation (using two selected LAB cultures) of immature tomatoes, which is an un-valorised resource of industrial tomato production, in order to support circular economy-oriented innovation.

## 2. Materials and Methods

### 2.1. Plant Material

An immature industrial tomato variety H1015 (unripe green; stage 1 fruit according to the USDA colour classification requirements) was collected, as per the process previously described by Pereira et al. (2021) [19]. A fruit selection [19] and preparation were performed upon arrival in the laboratory, and selected fruits were kept at −20 °C (Cryocell Aralab, Rio de Mouro, Portugal) until analysis.

### 2.2. Preparation of LAB Cultures

Two strains of pure LAB cultures, one isolated from immature tomato microbiota (*Lactiplantibacillus plantarum*; LAB97) and another from the INIAV IP (National Institute for Agrarian and Veterinarian Research IP) bacterial strain collection (*Weissella paramesenteroides*; C1090), with known resistance to low pH and bile salts [19], were tested for probiotic potential and as starters in the fermentation of the immature tomato pulp. The strains were activated by double culture in de Man, Rogosa, and Sharpe (MRS) broth (Biokar Diagnostics, Allone, France), and were incubated at 30 °C for 24 h. The suspensions were dispersed in a 0.85% NaCl solution (Sigma-Aldrich, St. Louis, MO, USA), which was adjusted to a MacFarland Standard turbidity of 5, and then distributed into fermentation flasks to achieve a final concentration of 8 log_10_ CFU/mL.

### 2.3. In Vitro Gastrointestinal Digestion Model

The tolerance to the in-vitro-simulated gastrointestinal conditions of the selected LAB strains was assessed according to process detailed by Fernández De Palencia et al. (2008) [20] and Moreira (2013) [21], with some modifications. The analysis was conducted in a metabolic water bath (Dubnoff MA-095, Marconi, Piracicaba, Brazil) at 37 °C to simulate the human body’s temperature and mechanical agitation (50 rpm) in order to simulate peristaltic bowel movements, with intensities similar to those achieved in the digestive tract. The experiments were performed in triplicate; the three independent cultures of each bacterium were analysed as follows.

Each previously propagated LAB strain (Section 2.2) was centrifuged three times (5000 rpm for 15 min at 4 °C), and the corresponding pellet was resuspended in 20 mL of phosphate buffer saline solution (PBS; 8.0 g/L NaCl, 0.2 g/L KH_2_PO_4_, 1.15 g/L Na_2_HPO_4_ at pH 7.2). After the final centrifugation, the pellets were resuspended in 35 mL of sterile electrolyte solution, thereby simulating salivary fluid (SSF; 6.2 g/L NaCl, 2.2 g/L KCl, 0.22 g/L CaCl_2_, 1.2 g/L NaHCO_3_, at pH 6.2). The solution was thoroughly mixed with a vortex mixer (Prolab, São Paulo, Brazil) before being transferred to the sterile 100 mL flasks. The flasks were placed in a 37 °C water bath for 20 min before starting the first digestion step (t_0_).

*Digestion procedures:* In the oral phase, 5 mL of SSF [with salivary lysozyme (0.01% *m/v*; Sigma-Aldrich, St. Louis, MO, USA)] was added to 35 mL of each strain in an MRS broth and then incubated for 5 min at 37 °C under stirring (50 rpm). Three mL of simulated gastric fluid (SGF; 6.2 g/L NaCl, 2.2 g/L KCl, 0.22 g/L CaCl_2_, 1.2 g/L NaHCO_3_, and pepsin (0.03% *m/v*; Sigma-Aldrich, St. Louis, MO, USA)) was added to the resulting suspensions from the oral phase in order to simulate the gastric conditions. The oral bolus was incubated at 37 °C under stirring (50 rpm) at pH 2.5 (adjusted with 1 M HCl) for 1 h (t_S_). Afterwards, the pH of the gastric chyme was adjusted to 6.5 by adding NaOH (1 M) and 4 mL of simulated intestinal fluid (SIF; 5 g/L NaCl, 0.6 g/L KCl, 0.3 g/L CaCl_2_, pancreatin (0.1% *m/v*; Sigma-Aldrich, St. Louis, MO, USA) and bile salts (0.45% *m/v*; Sigma-Aldrich, St. Louis, MO, USA)), which was added to the samples. The mixed suspensions were incubated at 37 °C under stirring (50 rpm) at pH 6.5 for 2 h (t_F_). Alongside the cell suspensions, each strain was kept in PBS (same proportions) and subjected to the same procedures, except for the addition of enzymes, bile salts, and pH adjustments as a control (Id: CTR). The LAB82 strain (*Leuconostoc citreum*; autochthonous to immature tomato) was used as a reference due to its known sensitivity to low pH conditions and due to the presence of bile salts [19].

*Cell survival analysis:* In the initial cell suspension (t_0_), and at the end of each digestion step (t_S_ and t_F_ for gastric and intestinal steps, respectively), the cell viability was monitored by LAB plate counts (see Section 2.5.1).

### 2.4. Immature Tomato Pulp Fermentation

*Immature tomato pulp preparation:* Pulp processing was conducted in a sanitised room using sanitised apparatuses in order to prevent contamination during processing. Immature tomato pulps were prepared from previously tawed fruit at 5 °C (24 h). The immature tomatoes were homogenised with 1.5% NaCl in a Robot Thermomix (Vorwerk, Germany), set at maximum speed for 1 min. After homogenisation, the pulps (500 mL) were distributed (NU-201 4 ft Laminar Flow Hood, NuAire) into 1 L Schoot flasks in order to set up 4 sample types: two individually inoculated samples with starters (Id: LAB97 and C1090) and two types of control samples (Id: CTR-TT and CTR-NTT). In the samples to be inoculated, and also in one of the control samples (CTR-TT), the pulps were thermally treated (110 °C/2 min) to minimise the influence of the microbiota that were present in the raw material.

*Fermentation conditions*: Only the LAB97 and C1090 samples were individually inoculated after the pulp’s thermal treatment. Inoculation was standardised at 8 log_10_ CFU/mL of pulp. Fermentation was carried out in capped 1-L Schoot flasks, in triplicates, at 25 °C (±1 °C) with continuous orbital stirring (100 rpm; Lab-Line Instruments, Melrose Park, IL, USA) for 14 d, and aliquots (ca. 20 g) were taken at regular intervals (0, 24, 72 h, 7th, and 14th days) in order to assess the following parameters: LAB and Y&M counts, titratable acidity, soluble solids content, CIELab colour, total phenolic content, antioxidant capacity, as well as the organic acids and phenolic profiles. A sensory evaluation by a panel was performed on the last day of storage (14th day).

### 2.5. Analytical Procedures

#### 2.5.1. Microbial Analysis and the Survival of Viable LAB Cells

According to ISO 15214 (1998) [22], viable LAB counts were determined using the pour plate method (MRS agar; Biokar Diagnostics, Allone, France). The LAB viability was monitored by LAB plate counts and expressed as log_10_ CFU/mL. The survival rate was calculated as follows:Survival rate (%) = (1 − (LAB counts at t_0_ − LAB counts at t_F_)/LAB counts at t_0_) × 100(1)

The yeast and mould (Y&M) counts (log_10_ CFU/mL) were performed according to ISO 21527-1 (2008) [23], using dichloran Rose-Bengal Chlortetracycline Agar (Biokar Diagnostics, Allonne, France).

#### 2.5.2. Titratable Acidity and Solid Soluble Content

The titratable acidity (TA) was determined according to NP-1421 (1977) [24]. The values were expressed as the mass equivalent (g) of lactic acid per 100 g of fresh weight (g LA/100 g FW). The solid soluble content (SSC) was determined using a digital refractometer (Atago Palette PR-201, Tokyo, Japan). The values were expressed as °Brix.

#### 2.5.3. Quantification of Total Phenolic Content and Antioxidant Capacity

Samples were homogenised with methanol (1:4, *w*:*v*) and left overnight at 5 °C. Homogenates were centrifuged at 29,000× *g* for 15 min at 5 °C (Sorvall RC5C, rotor SS34, Sorvall Instruments, Du Pont, Wilmington, DE, USA), and the clear supernatant (methanolic extract) was used for the total phenolic content, antioxidant capacity, and phenolic profile determinations.

The total phenolic content (TPC) was determined by the Folin–Ciocalteu method, as described by Alegria et al. (2021) [25], and the results were expressed in mg gallic acid equivalents, per 100 g of fresh weight (mg GAE/100 g FW).

The antioxidant capacity (AOx) was determined by the DPPH method, according to the procedures described by Brand-Williams et al. (1995) [26]. The results were then expressed in the μmol of Trolox equivalents, per 100 g of fresh weight (μmol TEAC/100 g FW).

#### 2.5.4. Organic Acids and Phenolic Profiles

Organic acid extraction was conducted by sample (3 g) homogenisation in a polytron (Ika, Ultra-Turrax T25, Staufen, Germany) with phosphate buffer (0.05 M, pH 2.8) till 13 mL of total volume, followed by 10 min of ultrasonication (Sotel Branson 2200 Ultrasonic Cleaner) and 20 min of centrifugation (Sigma, 2K15, Neustadt, Germany) at 4500 rpm and at 4 °C. The supernatant was filtered into an identified vial through a nylon syringe filter (Filter Lab, Barcelona, Spain) and was loaded into the high-performance liquid chromatography (HPLC) system. The organic acid profiles and quantifications were performed in a Waters HPLC system (Alliance 2690, 996 PDA and column thermostat JetStream 2 plus, Milford, MA, USA), which was coupled to a photodiode array detector (PDA), according to the process conducted by A. Panda et al. (2022) [27] with some modifications. Organic acids were separated in an ion-exclusion column (Rezex™ ROA, 300 × 7.8 mm, 8 μm particle size, Phenomenex, Torrance, CA, USA) at 25 °C in isocratic mode with a 0.01 M sulfuric acid mobile phase for 30 min at a flow rate of 0.5 mL/min and at 10 μL injection volumes. The identification of organic acids was made at 210 nm (254 nm for ascorbic acid) by comparing the organic acid standards’ retention time and UV spectrums. The peak areas were quantified and processed with the Empower Pro 2002 v.5.0 Software (Waters, Milford, MA, USA), which was conducted by comparison to the calibration of mix organic acids standards of citric, tartaric, succinic, malic, and formic acids (20 to 2000 µg/mL), oxalic acid (2 to 200 µg/mL), ascorbic acid (18 to 1300 µg/mL), lactic acid (18 to 1820 µg/mL), and acetic acid (10 to 760 µg/mL).

The phenolic profile was performed by HPLC-PDA in the same Waters HPLC system described above, and the phenolic compounds were separated at 25 °C in an RP column (Synergi Hydro, 250 × 4.6 mm, 4 μm particle size, Phenomenex Torrance, CA, USA), according to the process detailed by Petitjean-Freytet et al. (1991) [28] with minor modifications. The process of quantifying the phenolic compounds was based on a developed external standard curve using mixed standard solutions, ranging from 5 to 150 μg/mL. The hydroxycinnamic acids (chlorogenic, caffeic, coumaric, and ferulic) were integrated at 325 nm, the hydroxybenzoic acids (gallic, hydroxybenzoic, vanillic, and syringic), catechin, naringenin, and naringin were utilised at 280 nm, and the rutin, quercetin, and kaempferol were used at 340 nm.

The limit of detection (LD) and limit of quantification (LQ) for each compound were calculated based on the standard deviation (Sy) and the slope of the calibration curve (S), according to the following formulas:LD = 3.3 (Sy/S)(2)
LQ = 10 (Sy/S)(3)

The results were expressed in mg per 100 g of fresh weight.

#### 2.5.5. CIELab Colour

The colour of the immature green tomato pulps was evaluated in the CIELab system (Illuminant C) using the Minolta Chroma Meter CR-300 colourimeter (Osaka, Japan), and they calibrated with a white reference standard (L* = 97.10; a* = 0.19; b* = 1.95). The whiteness index (WI) and total colour difference (ΔE) were determined according to
(4)$$\mathrm{WI}\;=100-\sqrt{{\left(100-L^{*}\right)}^{2}+a^{*}{}^{2}+b^{*}{}^{2}}$$
(5)$$\Delta E\;=\sqrt{{\left(L^{*}-L_{0}^{*}\right)}^{2}+{\left(a^{*}-a_{0}^{*}\right)}^{2}+{\left(b^{*}-b_{0}^{*}\right)}^{2}}$$
respectively. In Equations (4) and (5), the L* values represent the luminosity of the samples (0—black to 100—white), and the a* and b* values indicate the variation of greenness to redness (−60 to +60) and blueness to yellowness (−60 to +60), respectively. Subscript “0” in Equation (5) refers to the initial colour parameter value that was assessed before fermentation.

#### 2.5.6. Sensory Analysis

A panel of 16 trained panellists conformed to ISO 8586-1 (1993) [29], as well as gathered adequate conditions in compliance with ISO 13299 (2016) [30]. The samples of the immature tomato fermentates were served at room temperature in glass cups (10 g each), which were marked with three-digit code numbers and were presented in a randomised order. The panellists were asked to identify and distinguish the sensory attributes, such as colour, flavour, consistency, and acceptance. Evaluations were scored based on a 9-point hedonic scale, with 1 representing the lowest score (disliked very much) and 9 the highest score (liked very much) [31].

### 2.6. Statistical Analysis

The statistical analysis was performed by using Statistica^TM^ v8.0 software from StatSoft [32]. Data were subjected to one-way or factorial ANOVA, and the means were compared using the Tukey HSD test (*p* = 0.05). The Pearson correlation coefficients were also determined between the studied responses.

## 3. Results and Discussion

### 3.1. Probiotic Potential Assessment of Two LAB Cultures

The probiotic characteristics of the LAB97 and C1090 strains were assessed using bacterial viability during an in vitro gastrointestinal digestion model simulation, with LAB82 provided as the reference of the low-tolerance strain. All bacterial strains maintained (*p* > 0.05) the initial counts (9 log_10_ CFU/mL) in the control condition (without the addition of enzymes, bile salts, and pH correction), thereby ensuring that the variations found during the successive stages of in-vitro-simulated digestive conditions were caused by them. The bacterial viability (LAB counts) for each strain that were tested during the different phases of gastrointestinal simulation is presented in Figure 1. As can be seen, each step of the simulated digestion process had a distinct impact on the survival of the tested strains. A one-hour incubation in an acidic stomach simulation (pH = 2.5; t_S_) significantly decreased the bacterial counts for all the LAB strains (ca. 3 log_10_ reduction for LAB97 and C1090 and ca. 3.5 log_10_ drop for LAB82).

In the following stage, both strains, LAB97 and C1090, were maintained at approximately 6 log_10_ CFU/mL after two hours of incubation (t_F_), thus showing their respective viability under the simulated intestinal conditions. On the contrary, the LAB82 microbial viability was significantly reduced (by about 1 log_10_ CFU/mL) when the pancreatic enzyme and bile salts were present in the intestinal phase, thereby demonstrating that this strain had a lower tolerance, reaching final counts of about 4 log_10_ CFU/mL.

The Sun et al. (2022) [33] and Yadav et al. (2022) [34] studies, for strains belonging to the *L. plantarum* (LAB97) and *W. paramesenteroides* (C1090) species, previously demonstrated a higher tolerance to unfavourable intestinal conditions than to adverse stomach conditions, particularly with respect to those regarding low acidity, which is consistent with our findings.

Strains LAB97 and C1090 exhibited comparable survival rates of around 67% throughout the simulated digestive process, whereas LAB82 showed a lower survival rate of 46%, thus confirming its previously low tolerance to acidic conditions and bile salts [19]. Furthermore, the LAB97 and C1090 strains met the viability criterion for probiotic potential by maintaining counts of ca. 6 log_10_ CFU/mL after the in vitro gastrointestinal digestion simulation [17], thus supporting the use of these strains as novel probiotic candidates.

Numerous bacterial species have been the focus of multiple investigations evaluating their probiotic properties—mainly those from the genera *Lactiplantibacillus*, *Weisella*, *Leuconostoc*, and *Pediococcus*. The probiotic potential of *W. paramesenteroides* was determined by Paula et al. (2014) [35], as well as by Sathyapriya and Anitha (2019) [36], based on their tolerance to simulated gastrointestinal tract conditions and due to the maintenance of viability. Moreover, numerous species from the genus *Lactiplantibacillus* have already been identified, demonstrating probiotic potential, so the LAB97 strain tolerance behaviour is not unexpected. Along with other probiotic traits, several *Lactobacillus plantarum* and *Lactobacillus lactis* are well recognised for their tolerance to unfavourable gastrointestinal conditions [37,38].

### 3.2. Quality Assessment of Fermented Immature Tomato Ingredients Inoculated with LAB Strains as Starters

#### 3.2.1. Microbial Growth, Acidification, and Soluble Solids Content during Fermentation

The LAB counts, as well as the TA and SSC changes, throughout the fermentation of the inoculated and non-inoculated immature tomato pulps are shown in Figure 2a,b, respectively.

As a general trend, the LAB counts of all samples increased during the fermentation period (first 72 h), reaching levels that ranged from 7.5 to 9.0 log_10_ CFU/ mL (Figure 2a). From this date onwards, the bacterial viability in both inoculated and non-inoculated samples was maintained, with no further variations to the end of the tested period. The LAB counts of the control samples on day 0 (<10 CFU/mL and 4 log_10_ CFU/mL for CTR-TT and CTR-NTT, respectively) were significantly lower than the inoculated samples, as expected. The LAB97 samples accounted for the highest LAB count value (ca. 9 log_10_ CFU/mL), which was achieved earlier at 24 h. Independent of the heat treatment or initial inoculum, the LAB growth after 72 h reached similar counts for all samples, demonstrating that immature tomato pulp is a favourable environment for LAB growth.

The Y&M growth (Appendix A, Appendix A) was similar amongst the samples, with no significant differences during the assessed period. However, the CTR-NTT samples had more significant counts on day 0 (2 log_10_ CFU/mL) than the others (<10 CFU/mL), as they were the only samples not heat-treated. As a result, the LAB group’s growth appeared to be mainly responsible for the differences in the parameter changes that were evaluated in the immature tomato fermentates.

Regarding acidity changes (Figure 2b), all of the samples showed an increase in TA values after 72 h of fermentation (ΔTA ≈ 1.3 g LA/100 g FW), except for LAB97, in which the significant TA variations occurred earlier (24 h) when compared with the remaining samples and which coincided with the period of exponential LAB97 growth (Figure 2a). A similar acidification delay in spontaneous fermentation, compared to controlled fermentation, was also reported in a study about the lactic acid fermentation of tomato pulp [39] and peppers [40].

The inoculated samples exhibited a faster and higher decrease rate in the SSC levels (ΔSSC ≈ 1.7 °Brix) than in the non-inoculated samples (ΔSSC ≈ 1.1 °Brix) (Figure 2b). This trend may be ascribed to the microbial consumption of soluble compounds (sugars) for their growth. These findings demonstrate that the chosen LAB strains could successfully develop on immature tomato pulps without nutrient supplementation or pH adjustment, thus potentially yielding a plant-based probiotic ingredient.

The TA increase found in all of the samples ascertained the progressive acidification of the fermented products throughout the tested period, and the SSC downward trend was concurrent with an increase in the substrate consumption for microbial growth. These variations were corroborated with the lactic fermentation occurrence in the immature tomato pulp. As mentioned, the LAB97 strain was isolated from immature tomato microbiota, which may account for its adaptability to the food matrix and in its high ability to ferment it. The LAB97 strain’s autochthonous nature and fermentative behaviour demonstrated its superior efficiency at fermenting immature tomato pulp when compared to the C1090 bacterial strain.

Similar acidification rates were found in tomato pulp (*Latilactobacillus sakei*) [39], unripe tomato pulp (a consortium of *L. plantarum*, *Leuconostoc mesenteroides*, and *Kluyveromyces marxianus*) [7], and tomato powder (*Pediococcus pentosaceus*) [41] when fermented by LAB starter cultures, with significant increases after 48 h of fermentation. Regarding the lactic fermentation of different plant products, including peppers [40] and several raw fruits and vegetables [42] (or mixtures of fruits and vegetables [43]), it was reported that there was a delay in the acidification process for spontaneous fermentation when compared to controlled fermentation. Di Cagno et al. [42] further stated that controlled fermentation is beneficial since it accelerates acidification, thus preventing the outgrowth of spoilage (or even pathogenic) microorganisms, which can compromise product safety and can increase the risk of undesirable sensory changes in the final product.

#### 3.2.2. Organic Acids Profile

From the organic acids profile, nine compounds were identified in the different samples (LAB97, C1090, CTR-NTT, CTR-TT); however, tartaric, succinic, ascorbic, malic, and formic acid contents were below the limit of quantification. In the different profiles, the lactic and acetic acids, which resulted from the fermentation process, and citric and oxalic acids, which were found in the fruit, were common between the samples. These were quantified, and their variations during the test period are presented in Table 1.

As expected, the predominant acid in the immature tomato pulp was citric acid (mean values of 940.2 mg/100 g at day 0) since it is the primary organic acid in raw tomato material [44]. No changes in oxalic acid content were registered during the tested period, regardless of sample type. Both citric and oxalic acids had a low correlation with TA (r = −0.44 and r = −0.03, respectively). On the other hand, the lactic acid content showed the strongest correlation with the TA (r = 0.92; *p* < 0.05), attesting to the lactic fermentation occurrence in the samples. Similar correlation values were found for the sweet potato fermentation, using *L. plantarum* strain as a starter into lacto-juice (r = 0.90; S. H. Panda and Ray, 2007) [45] and into pickles (r = 0.97; S. H. Panda et al., 2007) [46]. The increased concentration of the organic acids produced by LAB throughout the fermentation process, mainly lactic acid, was responsible for TA changes. The lactic and acetic acid concentrations increased along with fermentation, with the highest values (532.0 and 178.1 mg/100 g, respectively) found on the 14th day.

The LAB97 samples showed the most significant (*p* < 0.05) increases in lactic acid in the one hand, and minor increments of acetic acid in the other hand, compared to the remaining samples. LAB97’s greater efficiency could explain these variations, which prevented different microbiota from participating in the fermentation process. It should be noted that the acetic acid produced by acetic acid bacteria imparts distinctive aromas, with its presence typically associated with an unfavourable sour flavour in fermented products [47].

#### 3.2.3. Bioactive Composition (AOx, TPC, and Phenolic Profile)

Changes in all of the samples’ AOx and TPC, over time, followed a similar trend (Figure 3), with both parameters highly correlated (r = 0.93; *p* < 0.05). There was a nearly 2-fold increase (*p* < 0.05) between the 24 and 72 h periods, which was maintained until the end of the tested period (mean values of ca. 39.2 mg GAE/100 g FW and ca. 2518.9 μmol TEAC/100 g FW for TPC and AOx, respectively). The high correlation suggests that the fermentates’ antioxidant capacity is mainly attributed to the phenolic composition, as Torres et al. (2015) [48] reported. Furthermore, the significant increase in AOx and TPC matches the exponential LAB growth phase.

The literature concerning the variation of total phenolic content and antioxidant capacity during fermentation by different strains in various plant-based products depicts very diversified trends [49]. Depending on the strains used, there have been multiple references to increases, decreases, or to the maintenance of the initial contents throughout fermentation.

Yang et al. (2018) [50] showed that using *L. plantarum* in apple juice fermentation significantly decreased TPC during storage. The same trend was reported for the fermentation of papaya juice [51] and olive juice [52] using *L. plantarum* strains. Conversely, slight increases in TPC were reported for the tomato juice that was fermented by *L. plantarum* and *L. casei* [53]. For prickly pear juice (*Opuntia* sp., cv. Skinners Court), it was fermented by *L. fermentum*, in which no significant TPC variations were observed over 48 h [54]. In particular, the TPC values between 17 and 20 mg GAE/100 g and 19 and 37 mg GAE/100 g were reported in the tomato juice that was fermented by *L. casei* and *L. plantarum* [53], or by other LAB species [55], respectively, which is close to the range obtained in our samples.

Chen et al. (2018) [51] reported divergent variation trends in the AOx assessment of fermented papaya juice, depending on the strain employed as a starter: *L. acidophilus* caused a significant decrease in AOx, while *L. plantarum* led to a slight increase. Significant increases in AOx were noted during the fermentation of olives [52], tomato juice [53], and mulberry juice [56], which were fermented by different LAB strains, including *L. plantarum*. The AOx increase in fermented products can be ascribed to the accumulation of antioxidant compounds, including phenolic compounds [57].

The available information regarding the influence of phenolic compounds on LAB growth and viability is still scarce, and the metabolic pathways of phenolic compound biosynthesis or degradation by the LAB population have not yet been fully clarified [58]. Notwithstanding the microbial transformation and depolymerisation of high molecular weight phenolic compounds, this may account for some reported TPC increases during fermentation [59]. On the other hand, it has been found that phenolic degradation can be used as an adaptation mechanism to overcome the bacteriostatic effect exerted by high concentrations of phenolic compounds on the LAB population, ensuring further LAB growth. Hence, this mechanism might be accountable for the TPC decrease during fermentation [58]. To sum up, the interaction of the LAB population with food phenolics highly depends on the nature and amount of phenolic compounds that are present in the fermented feedstuff and in the strains involved in the fermentative process.

The evaluation of the phenolic profile (Appendix A, Appendix A) allowed the identification of nine phenolic compounds in all the samples: chlorogenic, hydroxybenzoic, vanillic, caffeic, syringic, coumaric and ferulic acids, as well as catechin, rutin, and naringin. However, it was impossible to quantify them accurately (given the quantification limits), which made the comparison of the contents between the samples unviable. The primary and most investigated phenolic compound in the mature tomatoes is chlorogenic acid [60], whilst the most prevalent flavonoids are glucosylated naringenin derivatives, as well as glucosylated quercetin, rutin, and kaempferol derivatives [61].

#### 3.2.4. CIELab Colour and Sensory Analysis

The fermented and unfermented samples’ appearance is shown in Figure 4.

The ΔE values express colour changes relative to the initial condition (unfermented samples), thereby indicating the colour stability of food products. Controlled and spontaneous fermentation influenced the ΔE in immature tomato pulps, as shown in Figure 5. The LAB97 samples showed significantly lower ΔE values (1.8, slight colour differences) when compared to the remaining sample types (≥3.3, very distinct colour differences), thus denoting better colourimetric stability. The ΔE values were highly correlated (r = 0.68; *p* < 0.05) to the whiteness index (WI) (data not shown). Therefore, the extent of colour changes has depended considerably on the fermented samples’ pulp darkening/browning changes. Enzymatic browning, because of polyphenol oxidases (PPO; EC 1.10.3.1), affects the colour quality of fruit and vegetable products. This enzymatic activity is hindered by an acidic environment (pH 3–4) [62]. Therefore, the LAB97 samples’ high colourimetric stability may be ascribed to fast acidification and to the subsequent enzymatic (PPO) inhibition effect, thus preventing browning reactions.

The sensory evaluation of the fermented products, as shown in Figure 6, was performed only on the 14th day by panellists who distinguished the sensory attributes of the samples’ colour, consistency, flavour, and global appreciation.

Despite the significant differences detected in the CIELab colour instrumental evaluation, the panellists could not distinguish the differences in colour between the inoculated and non-inoculated fermented samples (colour scores with non-significant differences, *p* > 0.05). A similar outcome was observed for the consistency attribute, with no differences noted between the samples. Thus, the samples’ colour and texture are unlikely to impact the fermentates’ sensory acceptance. On the other hand, the metabolic diversity of the several lactic acid fermentation strains may affect flavour development [63]. Regarding this attribute, the LAB97 sample was the most appreciated due to its pungent smell and balanced taste, whereas the flavour appreciation for CTR-TT, CTR-NTT, and C1090 was lower and similar (*p* > 0.05). During lactic acid fermentation, the lactic and acetic acids were the main organic acids produced, of which acetic acid was highly associated with an unpleasant vinegar flavour [64]. The LAB97 samples presented the lowest concentration of acetic acid after 14 days (5.5 mg/100 g; Table 1), which might contribute to its highest flavour score. Due to these factors, the LAB97 sample had the highest total acceptability, thus demonstrating the sensorial advantages of using the *L. plantarum* strain as a starter in the fermentation of immature tomatoes. Although LAB97’s overall acceptance was not very high (mean score ~5), it should be noted that these fermentates are ingredients to be used in food formulations rather than in a finished product. In addition, *L. plantarum* has been widely used as a starter culture in several fruit and vegetable fermentation processes, thereby contributing to the development of pleasant organoleptic properties (taste and texture) [65].

## 4. Conclusions

The *Lactobacillus plantarum* LAB97 starter culture demonstrated high lactic fermentation efficiency in immature tomato fruit pulps and also indicated a suitable tolerance to adverse conditions in the in vitro digestion model. Its use as a starter allows the valorisation of unripe tomato fruits by developing a healthy, appealing, probiotic candidate food ingredient, thereby adding economic value to food resources that used to be viewed as waste. Additional benefits may result from using this culture as an autochthonous starter; it allows the conservation of the indigenous biodiversity of food microbiomes as a more sustainable option. Furthermore, it is already adapted to the raw material and the processing environment, thus reducing the risk of contamination by pathogenic microorganisms during fermentation. Moreover, it can provide more consistent fermentation results, reducing the variability of the final product, which is essential for scalable commercial production.

## Figures and Tables

**Figure 1 foods-12-01532-f001:**
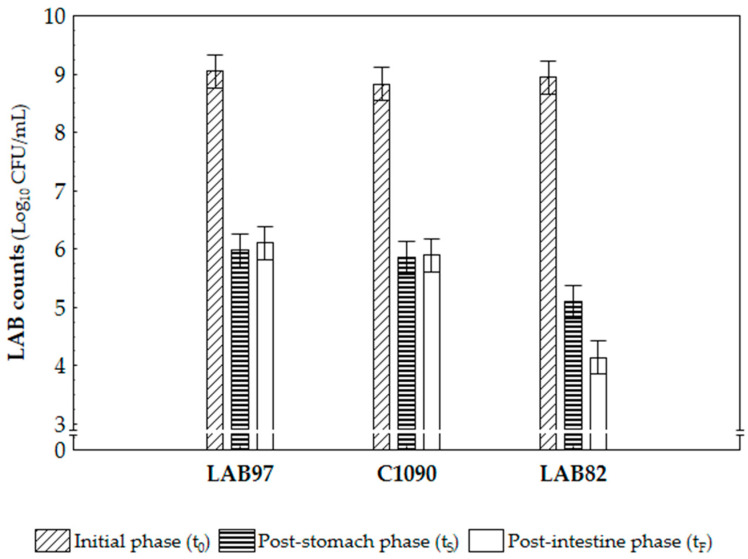
The LAB counts (log_10_ CFU/mL) for the LAB97, C1090, and LAB82 (as reference) strains under in-vitro-simulated gastrointestinal conditions. Bars represent the confidence intervals at 95%.

**Figure 2 foods-12-01532-f002:**
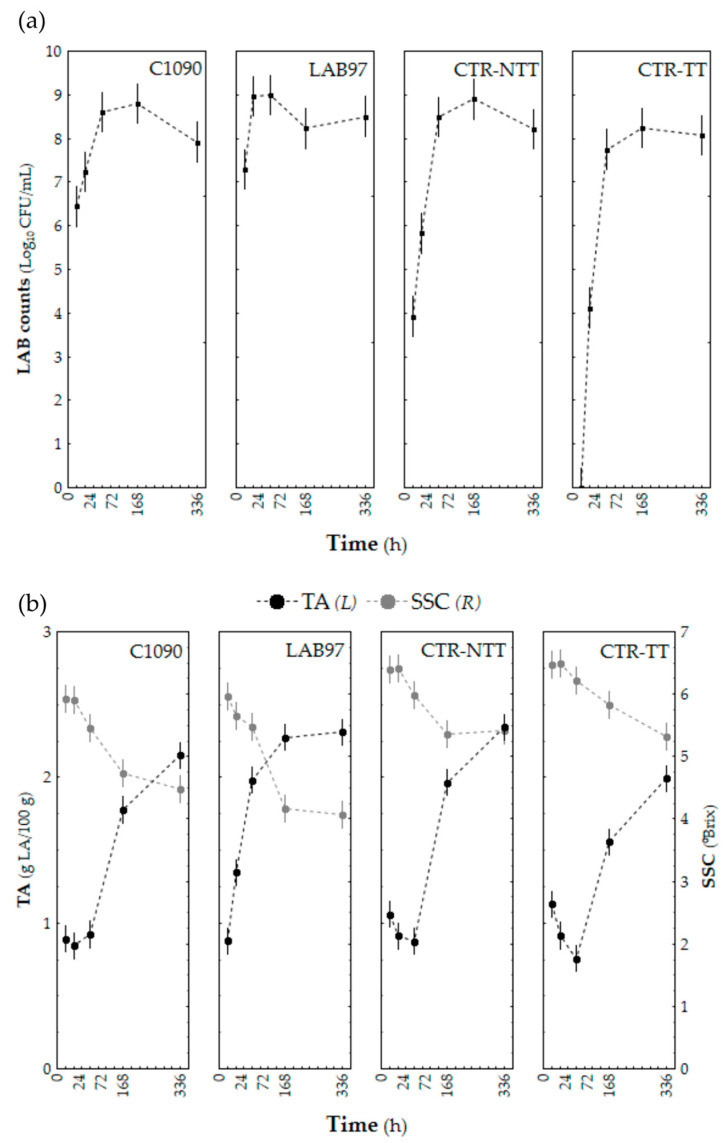
The LAB counts (log_10_ CFU/mL), (**a**) TA (g LA/100 g FW), and SSC (°Brix) (**b**) evolution throughout the lactic acid fermentation of immature tomato pulp samples that were inoculated with single LAB starter cultures (LAB97 and C1090) and non-inoculated samples (CTR-NTT and CTR-TT) for 14 days. Bars represent the confidence intervals at 95%.

**Figure 3 foods-12-01532-f003:**
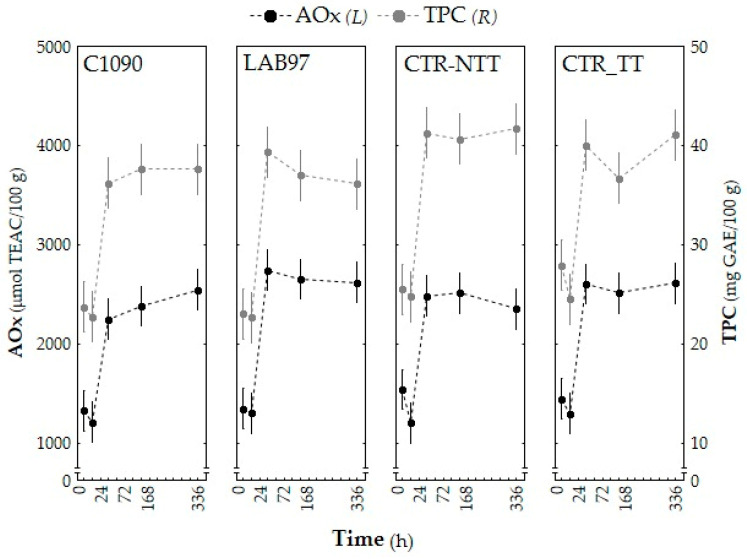
The AOx (µmol TEAC/100 g) and TPC (mg GAE/100 g) evolution throughout the lactic acid fermentation of immature tomato pulp samples, which were inoculated with single LAB starter cultures (LAB97 and C1090) and non-inoculated samples (CTR-NTT and CTR-TT) for 14 days. Bars represent the confidence intervals at 95%.

**Figure 4 foods-12-01532-f004:**
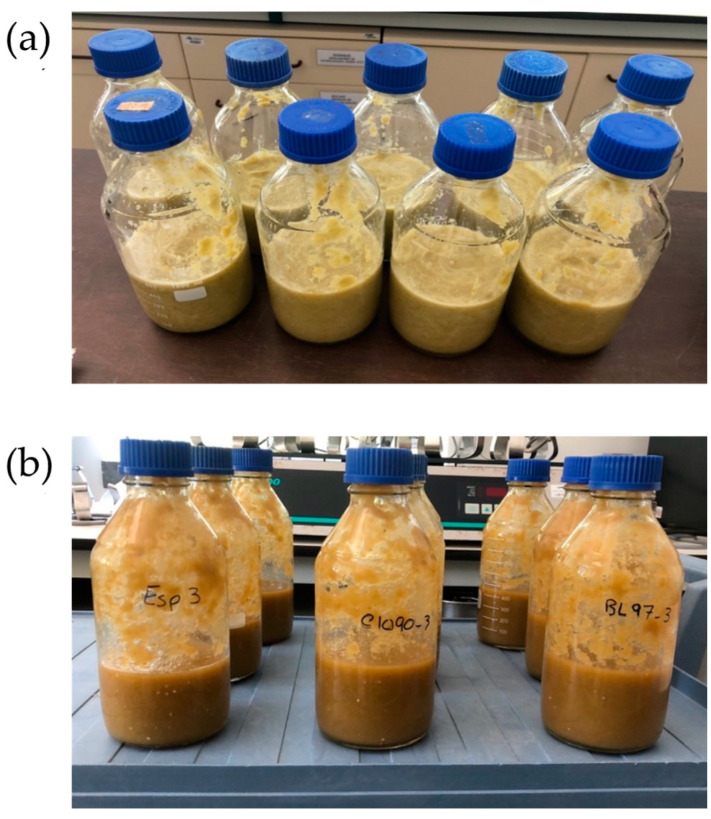
The appearance of unfermented (**a**) and fermented (**b**) immature tomato samples.

**Figure 5 foods-12-01532-f005:**
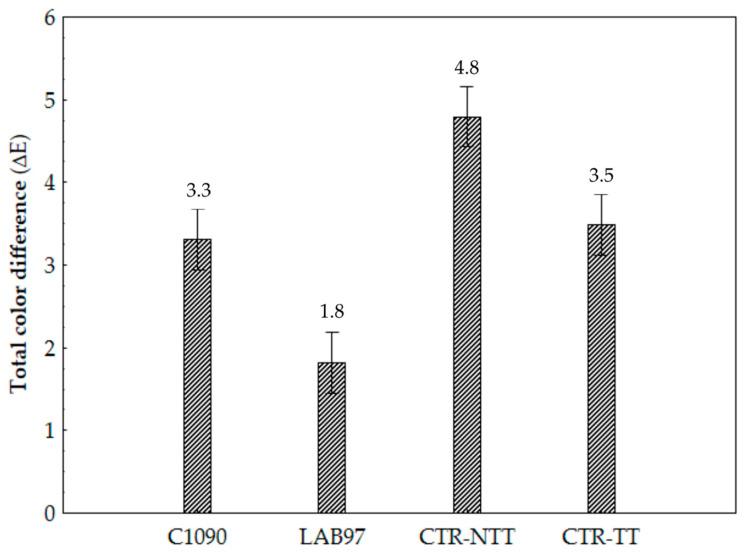
The average ΔE values of the immature tomato pulp samples that were inoculated with single LAB starter cultures (LAB97 and C1090) and non-inoculated samples (CTR-NTT and CTR-TT) after 14-day storage. Bars represent the confidence intervals at 95%.

**Figure 6 foods-12-01532-f006:**
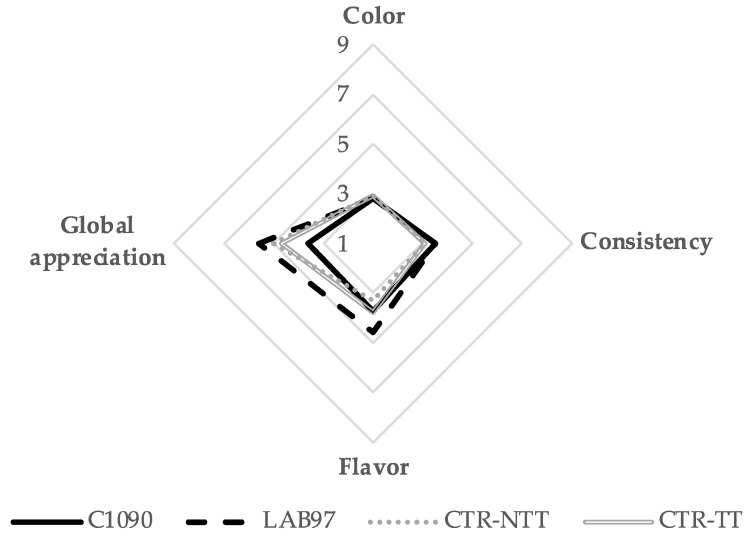
The average scores for the sensory attributes (colour, consistency, flavour, and global appreciation) of all the samples (LAB 97, C1090, CTR-NTT, and CTR-TT) on the 14th day.

**Table 1 foods-12-01532-t001:** The mean (and standard deviation) values of the organic acid contents (oxalic, citric, lactic, and acetic acids) for all samples (C1090, LAB97, CTR-NTT, and CTR-TT) throughout 14 days. For each column, the numbers followed by different letters are statistically different (Tukey HSD test, *p* = 0.05).

Sample	Time (h)	Oxalic Acid (mg/100 g)	Citric Acid (mg/100 g)	Lactic Acid (mg/100 g)	Acetic Acid (mg/100 g)
C1090	0	13.3 ± 1.3 ^bcd^	945.7 ± 56.0 ^def^	0.0 ± 0.0 ^a^	0.0 ± 0.0 ^a^
	24	12.9 ± 0.2 ^bcd^	886.0 ± 121.9 ^cde^	0.0 ± 0.0 ^a^	68.2 ± 13.6 ^abcd^
	72	12.3 ± 1.1 ^abcd^	711.1 ± 141.3 ^c^	212.9 ± 15.2 ^bc^	125.3 ± 4.2 ^bcde^
	168	14.4 ± 0.7 ^cde^	125.5 ± 24.2 ^a^	485.3 ± 49.6 ^ef^	141.6 ± 23.4 ^cdef^
	336	12.6 ± 0.9 ^abcd^	77.3 ± 23.1 ^a^	526.0 ± 110.2 ^efg^	157.6 ± 35.8 ^defg^
LAB97	0	12.0 ± 1.4 ^abcd^	907.0 ± 114.9 ^cdef^	0.0 ± 0.0 ^a^	0.0 ± 0.0 ^a^
	24	13.7 ± 0.2 ^bcde^	981.8 ± 17.3 ^ef^	258.6 ± 2.3 ^c^	0.0 ± 0.0 ^a^
	72	11.6 ± 1.2 ^ab^	863.1 ± 46.8 ^cde^	413.1 ± 13.1 ^de^	29.3 ± 1.8 ^ab^
	168	16.3 ± 0.9 ^e^	954.2 ± 28.6 ^def^	579.5 ± 25.2 ^fg^	64.6 ± 6.7 ^abcd^
	336	13.1 ± 0.6 ^bcd^	758.9 ± 17.5 ^cd^	525.1 ± 0.9 ^efg^	37.8 ± 5.5 ^abc^
CTR-NTT	0	12.0 ± 0.9 ^abcd^	973.7 ± 97.7 ^ef^	0.0 ± 0.0 ^a^	0.0 ± 0.0 ^a^
	24	12.1 ± 0.2 ^abcd^	1089.7 ± 2.5 ^f^	0.0 ± 0.0 ^a^	28.9 ± 5.6 ^ab^
	72	12.1 ± 0.0 ^abcd^	83.2 ± 16.3 ^a^	194.4 ± 6.6 ^bc^	191.9 ± 10.5 ^efg^
	168	10.1 ± 0.1 ^a^	33.2 ± 1.6 ^a^	424.4 ± 47.6 ^de^	212.9 ± 7.6 ^efg^
	336	11.7 ± 1.3 ^abc^	88.1 ± 8.3 ^a^	607.9 ± 12.0 ^g^	259.0 ± 13.5 ^g^
CTR-TT	0	13.8 ± 0.3 ^bcde^	934.3 ± 128.8 ^def^	0.0 ± 0.0 ^a^	0.0 ± 0.0 ^a^
	24	14.5 ± 0.2 ^de^	1017.8 ± 29.3 ^ef^	0.0 ± 0.0 ^a^	117.9 ± 6.6 ^bcde^
	72	13.8 ± 1.0 ^bcde^	498.0 ± 7.4 ^b^	122.4 ± 14.0 ^b^	148.0 ± 30.6 ^def^
	168	12.6 ± 1.3 ^abcd^	135.0 ± 24.6 ^a^	309.2 ± 65.0 ^cd^	241.1 ± 88.5 ^fg^
	336	11.8 ± 0.9 ^abc^	46.6 ± 4.5 ^a^	469.0 ± 91.1 ^ef^	258.8 ± 110.1 ^g^

## Data Availability

Not applicable.

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
