# Peer review of "Valorisation of Wasted Immature Tomato to Innovative Fermented Functional Foods"

_foods, 2023, doi:10.3390/foods12071532_

Round 1
Reviewer 1 Report
I have read and revised your paper, entitled "Valorisation of Wasted Immature Tomato to Innovative Fermented Functional Foods". I have found that you have presented a very complete work, but I have some doubts about the possible application of the outcomes of it. If immature tomatoes are discarded for consumption, why did the authors take this as potential functional food? Is it consumed in Portugal? Does it have a real possibility to me implemented? When talking about its potential use as an "ingredient" for a functional food, does the author have any idea that can be shared with the readers.
Some additional, few comments and corrections have been made along the text, which I am attaching for your use. I hope you find them useful.

Author Response
Reviewer #1
I have read and revised your paper, entitled "Valorisation of Wasted Immature Tomato to Innovative Fermented Functional Foods". I have found that you have presented a very complete work, but I have some doubts about the possible application of the outcomes of it. If immature tomatoes are discarded for consumption, why did the authors take this as potential functional food? Is it consumed in Portugal? Does it have a real possibility to me implemented? When talking about its potential use as an "ingredient" for a functional food, does the author have any idea that can be shared with the readers.
The tomato fruit for industrial production of tomato concentrate/paste (an industry of great national importance) is left in the field when failing to meet the required quality criteria at harvest (fully matured fruit [red color] and °Brix between 4 and 6, depending on cultivar). This work focused on the food valorisation of these fruits as biomass that consumed water, energy, and soil. The outlined strategy concerns using immature tomato fruit as a substrate for lactic fermentation to produce an ingredient for sauces’ formulation while considering the characteristics of fermented products, in this case, the probiotic potential.
This study is part of the OG GreenTaste project, under the Portuguese Rural Development 2014-2020 for Operational Groups program, with the objective of food valorisation under the circular economy concept. It is developed in close articulation with the Competence Centre for Industry Tomato (Centro de Competências para o Tomate de Indústria) to address the need to valorise a considerable volume of fruits that currently have no use.
On the other hand, the processing companies’ focus on product differentiation also has been increasing to strengthen this type of industry, previously focused on the supply of a single product. To this effect, a company will already benefit from this knowledge. The tomato processing companies members of the project consortium were also involved in broadening the range of products offered, having carried out tests with the developed ingredient. Following this work, sauce formulations have already been tested in which the developed fermented ingredient (65% proportion; LAB97) was included, showing promising results and will be the publication subject.
To clarify the outlined questions, we detailed this information in the introduction section (lines 58-68).
Some additional, few comments and corrections have been made along the text, which I am attaching for your use. I hope you find them useful (PDF). (we have identified the lines to which the reviewer commented on)
Line 46 - With the first sentence of the introduction, we aimed to highlight the global food waste estimates alongside the growing world population as drivers of the urgent concern to make food production more efficient, environmentally friendly, and sustainable. We rewrote the sentence to make it more straightforward (lines 45-47).
Line 61 – Thank you for pointing out this error. We corrected the unit (updated line 63). Estimates are 112 kilotonnes.
Lines 108-114 – As the reviewer duly noted, this is a passage from a previous paper already published by the same research team. This study is part of the same project (GreenTaste, as referenced in the funding section) where procedures and tomato cultivar have been maintained. We accommodated the reviewer's suggestion of including the reference and slightly modifying the text (Lines 113-118).
Line 117 – INIAV is the abbreviation of Instituto Nacional de Investigação Agrária e Veterinária, IP (National Institute for Agrarian and Veterinarian Research, IP), a Portuguese State Laboratory of the Ministry of Agriculture and Food. We included such information on lines 121-122.
Line 125 – We proceeded accordingly.
Line 133 – We corrected the section number (section 2.2)
Line 135 – We proceeded accordingly.
Line 140 – We proceeded accordingly.
Line 143 – From our experience, text editors usually correct us in this regard, starting a sentence with the number spelling instead of the number itself.
Line 144 – We are unsure of what the reviewer is asking. We have taken into account the pH of the gastric solution, adjusting the pH as stated in line 153 (to pH 2.5).
Line 163 – We proceeded accordingly.
Line 174 – We proceeded accordingly.
Line 176 – We proceeded accordingly.
Line 267 – Bluntly put, yes, conditions in the stomach (pH »2.5 ) are more acidic than in the intestine (pH »6.5), as generally accepted.
Line 293 – We have increased all graphs' sizes.
Lines 298, 304, 314, 315 – We proceeded accordingly.
Line 318 – In general, the controlled fermentation using endogenous strains adapted to the matrix as starters is faster than natural fermentation. However, we agree with the comment and have further added a reference to a study conducted on tomatoes.
Lines 323-324 – The use of this strategy in the food valorisation of these fruits is supported by the fact that immature tomato pulp is a suitable matrix for lactic fermentation with the selected endogenous LAB strains, as demonstrated, with no significant inputs to the fermentation factors (supplementation) and yielding a probiotic fermented pulp. Due to the fermented product's excessive acidification, we foresee its use as an ingredient rather than a finished product, specifically for developing sauces with probiotic characteristics.
Lines 334 and 335 – We proceeded accordingly.
Line 337 – We thank the reviewer for pointing this out to us. We corrected the text accordingly.
Line 344 – We agree with the reviewer that the use of "only" is reductive. We have detailed the explanation in lines 370-375.
Line 362 – Yes, during the fermentation process, lactic and acetic acids levels will increase. According to multiple studies, lactic and acetic acids are the main organic acids produced during the lactic fermentation of green tomato fruits. Their relative content is influenced by several factors, including the type of microorganisms involved, fermentation conditions (such as temperature, pH, and salt concentration), and tomato fruit maturity.
Line 367 – Thank you for your comment and the opportunity to explain our thought. The levels of organic acids can differently affect the sensory profile of a product. In this case, acetic acid can contribute to a pungent aroma, while lactic acid can contribute to a smooth and creamy texture. Nonetheless, the levels of each individual organic acid do not necessarily determine the full extent of their sensory impact. For that reason, it should be noticed that the significantly lower amounts of acetic acid, regarding lactic acid, may affect the flavour of the fermentates.
Line 379 – We proceeded accordingly.
Line 423 – We understand your comment; however, while ripe tomatoes are a good source of lycopene, green tomatoes contain only small amounts of this phytochemical. The low levels of lycopene in immature green tomato justify the option of not quantifying this response as its contribution to the overall antioxidant capacity would be quite limited (if existing) in comparison to e.g., flavonoids or phenolic acids.
Line 442 – Thank you for pointing this out to us. We have added the necessary information in the material and methods section (lines 257-263).

Reviewer 2 Report
1. What’s the activity of digestive enzymes (lysozyme, pepsin, pancreatin) used in vitro gastrointestinal digestion model?
2. Fermentation conditions: Please describe the fermentation conditions in details.
3. Why didn't the authors analyze the amount of carotenoids in immature tomatoes before and after lactic fermentation?
4. I am concerned about toxic substances in immature tomato. Did the authors measure the level of solanine in immature tomatoes before and after lactic fermentation?
5. Please include visual images depicting immature tomato pulp before and after lactic fermentation.
Author Response
Reviewer #2
- What’s the activity of digestive enzymes (lysozyme, pepsin, pancreatin) used in vitro gastrointestinal digestion model?
In the in vitro trial simulating the gastrointestinal digestion model, the enzyme activities were, by calculation, as follows: lysozyme – 75 U/mL; pepsin – 2000 U/mL; pancreatin – 100 U/mL.
- Fermentation conditions: Please describe the fermentation conditions in details.
We added further information in the section describing fermentation conditions (lines 177 – 185).
- Why didn't the authors analyze the amount of carotenoids in immature tomatoes before and after lactic fermentation?
In line with a question by one of the reviewers, green tomatoes contain small amounts of lycopene, the major tomato carotenoid. The low levels of this phytochemical in immature green tomato justify the option of not quantifying this response as its contribution to the overall antioxidant capacity would be quite limited (if existing) in comparison to e.g., flavonoids or phenolic acids.
- I am concerned about toxic substances in immature tomato. Did the authors measure the level of solanine in immature tomatoes before and after lactic fermentation?
In the project where this study is included, another research team dedicated to the issue of the alkaloids’ presence, such as solanine and tomatine, verified the low concentration of these compounds in both raw and fermented immature green tomatoes (cf. Simões et al., 2021). Although this quantification was not targeted in the present study, evidence suggests that the presence of these substances is not worrying and will be considered in the validation of the ingredient prototype. Still with this concern, in previous work (cf. Pereira et al., 2021), we found that the strains under study show tolerance to solanine. This result suggests that the strains may metabolise these compounds, reducing the potential associated risk. Nonetheless, this hypothesis must be tested in future research.
Simões, S.; Santos, R.; Bento-Silva, A.; Santos, M.V.; Mota, M.; Duarte, N.; Sousa, I.; Raymundo, A.; Prista, C. Improving nutritional quality of unripe tomato through fermentation by a consortium of yeast and lactic acid bacteria. J Sci Food Agric, 2021, 1–8. doi: 10.1002/jsfa.11476.
Pereira, N.; Alegria, C.; Aleixo, C.; Martins, P.; Gonçalves, E.M.; Abreu, M. Selection of autochthonous LAB strains of unripe green tomato towards the production of highly nutritious lacto-fermented ingredients. Foods, 2021, 10(12). doi: 10.3390/foods10122916
- Please include visual images depicting immature tomato pulp before and after lactic fermentation.
Thank you for the suggestion. We added figure 4 (line 456) to include photographic records of the immature green tomato pulps before and after fermentation.

Reviewer 3 Report
MS foods-2303169: Valorization of Wasted Immature Tomato to Innovative Fermented Functional Foods
Overall appreciation:
The authors assessed the probiotic properties of Lactiplantibacillus plantarum (LAB97, isolated from immature tomato microbiota) and Weissella paramesenteroides (C1090, from INIAV collection) through in vitro gastrointestinal digestion simulation. Then they assessed the LAB starters’ fermentative ability using partially decontaminated (110 °C/2 min) immature tomato pulp. The main quality parameters were assessed on day 14th.The whole work was acceptable but the sensory analysis is not well described and may be not well conducted. If hedonic test is conducted, we need at least 60 subject to do the analysis. If expert panelist is involved, the number fixed in standard is 12 (number of expert). Some issues with the methodology and results should be revised: sensory analysis, phenolic profile (chromatogram should be provided and if components are no quantified, this section should be removed from te manuscript. Why did the author choose to analyses phenolic compounds with are in the quantification limits), and didn’t analyze lycopene the main bioactive carotenoid of the tomato and other bioactive such as folate and vitamin C?!Besides the count of total viable LAB using bacterial in vitro gastrointestinal digestion model simulation, is not enough to use the term fermented functional foods, other analysis should be performed ad safety should be confirmed.
The title
Title: valorization with ‘z’
Abstract
-Lines 23 : …. up to 25 tonnes/ha:
This sentence should be in introduction with corresponding year, place, region? and reference
-line 32:
all samples (triplicates) were tested
(triplicates should be mentioned in material and method section.
Material and methods:
-tomatoes were selected: selected according to what
-The method of organic acid extraction should be added in material and method section. Only chromatographic analysis of fatty acids was described
-Line 194: the method of extraction of TPC (solvent of extraction, m/v, temperature and time) should be briefly and clearly described.
-line 2033, L,a,b and L0,a0,b0 should be defined.
-all equation should be numbered.
-171, fermentation was made at which temperature from d 0 to d 14, ho
Results and discussion
-the quality of fi 1, 2, 3, 4 and 5 should be improved, resolution, and character are small and not clearly readable
-the sensory analysis is not well described, which type of tests was used, hedonic, quantitative sensory profile ? description of the assay is incomplete and not clear. Why the author chose to do analyze on day 14th, it will be better to analyze on d0 and d 14
-table 1 : standard deviation should be added to the average values.
-Is the product is safe, what about the isolated LAB safety. Author mention reference 1 related to a previous identification
-the score attributed to the color, consistency and flavor are low, so how was the global appreciation is high (5/9) whereas all attributes were ranked lower

Author Response
Reviewer #3
Overall appreciation:
The authors assessed the probiotic properties of Lactiplantibacillus plantarum (LAB97, isolated from immature tomato microbiota) and Weissella paramesenteroides (C1090, from INIAV collection) through in vitro gastrointestinal digestion simulation. Then they assessed the LAB starters’ fermentative ability using partially decontaminated (110 °C/2 min) immature tomato pulp. The main quality parameters were assessed on day 14th.The whole work was acceptable but the sensory analysis is not well described and may be not well conducted. If hedonic test is conducted, we need at least 60 subject to do the analysis. If expert panelist is involved, the number fixed in standard is 12 (number of expert). Some issues with the methodology and results should be revised: sensory analysis, phenolic profile (chromatogram should be provided and if components are no quantified, this section should be removed from the manuscript. Why did the author choose to analyses phenolic compounds with are in the quantification limits), and didn’t analyze lycopene the main bioactive carotenoid of the tomato and other bioactive such as folate and vitamin C?!Besides the count of total viable LAB using bacterial in vitro gastrointestinal digestion model simulation, is not enough to use the term fermented functional foods, other analysis should be performed ad safety should be confirmed.
Addressing the reviewer's concern about the sensory evaluation, we point out that a trained panel of 16 panellists conducted the hedonic test. Further details were added to the sensory evaluation description (lines 257-263).
Concerning the provision of the phenolic profile chromatograms, we have added them to the supplementary material and referenced them in the discussion (line 446).
Concerning the choice of analysing phenolics instead of e.g., lycopene, the amount of that phytochemical in immature green tomatoes may be considered marginal when compared to e.g., flavonoids and phenolic acids, major contributors to immature green tomato antioxidant capacity. Further, it is hypothesised in other studies that LAB strains may contribute to the increase of phenolic compounds during the fermentation process, as supported by our results. Moreover, we stress that ascorbic acid content was also evaluated during the tested period; however, with contents below the quantification limit.
We agree with the reviewer that other analyses should be performed to fully understand the usability of this fermentate as a probiotic ingredient. This is the subject of the team’s current research. Nonetheless, a major step in considering a strain as a probiotic candidate is its viability along the gastrointestinal digestion (≥ 6-7 log CFU/mL), the present study’s focus. Other functional, safety and antimicrobial characteristics are to be explored in both the strains and during the ingredient and product shelf-life.
The title
Title: valorization with ‘z’
We thank the reviewer for the indication. Surely you must understand that as non-native European English speakers, we tend to write in British-English. Such conformity with American- or British-English is expected at a later stage of the revision process.
Abstract
-Lines 23: …. up to 25 tonnes/ha:
This sentence should be in introduction with corresponding year, place, region? and reference
We thank the reviewer and detailed the requested information in the introduction (lines 61-64).
-line 32: all samples (triplicates) were tested
(triplicates should be mentioned in material and method section.
We proceeded accordingly (e.g., lines 136, 179).
Material and methods:
-tomatoes were selected: selected according to what
Fruits were selected to remove damaged units and maintain colour uniformity (green fruit; stage 1 fruit according to the USDA colour classification requirements). This description was altered according to reviewer 1, referencing the information.
-The method of organic acid extraction should be added in material and method section. Only chromatographic analysis of fatty acids was described.
We thank the reviewer for pointing this out to us. The description of the extraction of organic acids was added (lines 214-220) prior to the chromatographic analysis of organic acids.
-Line 194: the method of extraction of TPC (solvent of extraction, m/v, temperature and time) should be briefly and clearly described.
We thank the reviewer. The description of the extraction was added (lines 202-206).
-line 230, L,a,b and L0,a0,b0 should be defined.
We proceeded accordingly and added the requested information in lines 252-255.
-all equation should be numbered.
We proceeded accordingly.
-171, fermentation was made at which temperature from d 0 to d 14, ho
As stated in the material and methods section (line 179), the fermentation was conducted at 25 °C.
Results and discussion
-the quality of fig 1, 2, 3, 4 and 5 should be improved, resolution, and character are small and not clearly readable
We have increased the size/resolution of the figures.
-the sensory analysis is not well described, which type of tests was used, hedonic, quantitative sensory profile ? description of the assay is incomplete and not clear. Why the author chose to do analyze on day 14th, it will be better to analyze on d0 and d 14
As pointed out by the reviewer, we have further detailed the description of the sensory evaluation (lines 257-263).
Indeed, the analysis at the reviewer’s indicated moments, day 0 and day 14, could be an interesting option to verify the profound changes induced by the fermentation. We selected the end of the tested period (day 14) because we considered this to be the moment when the ingredient’s sensory profile would be developed/stabilised, particularly, in what concerns flavour.
-table 1: standard deviation should be added to the average values.
We proceeded accordingly; however, we had omitted the standard deviation since the significance is shown.
-Is the product is safe, what about the isolated LAB safety. Author mention reference 1 related to a previous identification
We are sorry, but we do not quite understand your question. This work did not address safety issues related to the LAB strains under test. Only the strains' tolerance to gastrointestinal digestion conditions was assessed.
-the score attributed to the color, consistency and flavor are low, so how was the global appreciation is high (5/9) whereas all attributes were ranked lower
The panellists were informed that the product under evaluation was a potential ingredient, not a final product. The appreciation of colour and consistency was not highly valued as an ingredient. Then, flavour appreciation had a greater impact on the overall appreciation of the samples, justifying the sample scoring.

Round 2
Reviewer 2 Report
1. The total phenol content cannot indicate changes in carotenoid content because carotenoids do not belong to the phenol category.
2. When tomatoes are used in functional foods, consumers are highly concerned about the functional ingredient content, particularly carotenoids.
Author Response
Responses to the Reviewers’ comments (Round 2)
The responses are given below each reviewer’s comment.
Reviewer #2
- The total phenol content cannot indicate changes in carotenoid content because carotenoids do not belong to the phenol category.
As the referee’s comment states, total carotenoids are not related to total phenolic content. We never made such a statement in the text of the article or in the responses given at the previous round of the review process.
We might not have been clear in the previous round when mentioning that the lycopene content in immature green tomato fruit is very low as lycopene biosynthesis occurs during fruit maturation. This justifies our choice to assess the total phenolic content and determine the antioxidant capacity (which are highly correlated) in immature green tomatoes.
- When tomatoes are used in functional foods, consumers are highly concerned about the functional ingredient content, particularly carotenoids.
The relevant phytochemical composition - phenolics, carotenoids and other antioxidant compounds - contributes to defining an ingredient as functional, alongside with other characteristics, e.g. probiotic potential. The added value of the fermented ingredient in this research relies more upon the fact that it is a candidate probiotic than on the contribution of its antioxidant composition.
